# Engineering and Implementation of Synthetic Molecular Tools in the Basidiomycete Fungus *Ustilago maydis*

**DOI:** 10.3390/jof9040480

**Published:** 2023-04-17

**Authors:** Nicole Heucken, Kun Tang, Lisa Hüsemann, Natascha Heßler, Kira Müntjes, Michael Feldbrügge, Vera Göhre, Matias D. Zurbriggen

**Affiliations:** 1Institute of Synthetic Biology, University of Düsseldorf, 40225 Düsseldorf, Germany; 2Institute of Microbiology, University of Düsseldorf, 40225 Düsseldorf, Germany; 3CEPLAS—Cluster of Excellence on Plant Sciences, 40225 Düsseldorf, Germany

**Keywords:** synthetic biology molecular tools, ratiometric luminescence reporters, quantitative gene expression readout, bidirectional promoters, luciferase

## Abstract

The basidiomycete *Ustilago maydis* is a well-characterized model organism for studying pathogen–host interactions and of great interest for a broad spectrum of biotechnological applications. To facilitate research and enable applications, in this study, three luminescence-based and one enzymatic quantitative reporter were implemented and characterized. Several dual-reporter constructs were generated for ratiometric normalization that can be used as a fast-screening platform for reporter gene expression, applicable to *in vitro* and *in vivo* detection. Furthermore, synthetic bidirectional promoters that enable bicisitronic expression for gene expression studies and engineering strategies were constructed and implemented. These noninvasive, quantitative reporters and expression tools will significantly widen the application range of biotechnology in *U. maydis* and enable the in planta detection of fungal infection.

## 1. Introduction

The basidiomycete *Ustilago maydis*, which causes corn smut disease, is a well-characterized model organism for studying pathogen–host interactions [1,2,3,4,5,6]. In addition, its completely sequenced genome [7] increases the potential for biotechnology applications [1,3], as *U. maydis* is exceptionally well suited for genetic modification [8,9]. *U. maydis* has been engineered for the production of a number of biotechnologically relevant compounds such as terpenoids [10,11]. The strains can be cultured in relatively cheap media and the system is scalable to bioreactor production mass to withstand the application of the investigative methods and reagents.

*U. maydis* has several advantages in laboratory handling: (1) It can grow as haploid, yeast-like single cells in liquid culture and replicates through germination [1]. Applying glucose as the sole carbon source, the replication time is less than two hours. (2) These haploid cells are also very robust in high osmotic media and seawater and, compared to filamentous fungi, they are easier to cultivate in culture including large-scale culture in bioreactors [12]. (3) The cells can be engineered using effective homologous recombination yielding stable insertions, [8] and recently, CRISPR-Cas9-based genome editing technologies have been implemented [13]. (4) The lifestyle of *U. maydis* can be readily switched from yeast to hyphal growth in a genetically engineered strain by switching the nitrogen source in the medium [14], and solopathogenic strains enable the rapid analysis of infection [7]. 

A set of molecular tools is available for the genetic manipulation of *U. maydis*, including constitutive promoters, fluorescent reporters, and epitope tags [9]. In this context, 2A peptides for polycistronic expression in *U. maydis* were recently implemented [15]. However, reporters and other expression tools for the ratiometric, quantitative readout of gene expression are still lacking. In other organisms, synthetic molecular tools such as quantitative luminescent reporters, internal ribosome entry site (IRES) sequences, and bidirectional promoters (BPs) are well established. For example, during plant infection, bacterial proliferation can be quantified in a non-invasive manner in strains expressing the bacterial LUX-operon [16]. In plant cells, dual-luciferase assays with different substrates are regularly employed for monitoring gene expression and normalization [17,18]. IRES sequences are also commonly used in mammalian cell lines for bicistronic expression [19], and furthermore, bidirectional promoters were demonstrated to allow dual-gene expression, leading to robust, stable, and compact gene expression constructs [20]. 

Our aim is to customize, characterize, and implement a set of quantitative luminescence and enzymatic reporters in *U. maydis* which enable the ratiometric, quantitative monitoring of (inducible) gene expression and protein levels. In addition, two synthetic bidirectional promoters were engineered for bicistronic expression control to facilitate and expand engineering capabilities in the basidiomycete. In a proof-of-principle application, a set up for the non-invasive monitoring of the infection process, namely *U. maydis* infection of a plant leaf, was developed. In addition, this approach will significantly widen the application range of biotechnology in *U. maydis*.

## 2. Materials and Methods 

### 2.1. Plasmid Generation

The plasmids and oligonucleotides used herein are listed and described in Appendix A. Plasmids were generated with AQUA cloning or Gibson Assembly [21,22] and correctness was confirmed by sequencing. Transformation and plasmid isolation were performed in *Escherichia coli* TOP10 using standard techniques. Heterologous genes were codon-optimized for expression in *U. maydis*. 

### 2.2. Strain Generation and Growth Conditions

The *U. maydis* strains used in this work (Appendix A) derive from the lab strains AB33 [14], SG200 [7], and SG200-pit1∆ [23]. The cells were grown in complete medium (CM: 0.25% *w/v* cas-amino acids (Difco), 0.1% *w/v* yeast extract (Difco), 1.0% *v/v* vitamin solution, and 6.25% *v/v* salt solution from Holliday [24], 0.05% *w/v* deoxyribonucleic acid from herring sperm, and 0.15% *w/v* NH_4_NO_3_ adjusted to pH 7.0 with NaOH (Sigma Aldrich, Darmstadt, Germany) and supplemented with 1% glucose. The transformation of *U. maydis* protoplasts followed the protocol from [25]. The constructs were integrated using homologous recombination into the *pep4* or *upp3* locus, and the transformants were selected on the corresponding antibiotics (200 μg/mL hygromycin, 150 μg/mL nourseothricin, 2 μg/mL carboxin (Sigma Aldrich, Darmstadt, Germany)). 

For fluorescence analysis with plate readers, the strains were grown overnight in CM-glucose, before shifting the cultures to nitrate minimal medium (NM)-glucose for the induction of filamentous growth in AB33. Hyphal growth was induced by changing the media to nitrate minimal medium (NM) supplemented with 1% glucose or arabinose. All strains were confirmed using Southern blot analysis [8] or genotyping PCR.

### 2.3. Luminescence Determination

Luminescence was determined using a Berthold Technologies Centro XS3 LB960 Microplate luminometer. For all reporter gene assays, 5 mL CM-glucose cultures were inoculated with the reporter-gene-containing strains and incubated on a rotating wheel for 24 h at 28 °C. The cultures were adjusted to an OD_600_ of 0.5 in a total volume of 3 mL of CM-glucose. For cell lysis, cell pellets were resuspended in 1 mL of lysis buffer supplemented with protease inhibitors (50 mM Tris-HCl pH 7.4, 150 mM NaCl, 0.5 mM EDTA pH 8.0, 0.5% Nonident-P-40, 1 mM PMSF, 1 mM DTT, 2.5 mM benzamidine, and 200 μL of complete protease inhibitor cocktail (Sigma Aldrich, Darmstadt, Germany)). Afterwards, 2 scoops (~400 μL) of glass beads were added and the samples were incubated at 4 °C and 1500 rpm for 20 min and transferred into 1.5 mL reaction tubes.

For the measurement, the cell lysate and the supernatant were transferred to white 96-well plates and the substrates were freshly prepared and added before the measurement. Luminescence was determined by pipetting 80 μL of whole culture, diluted lysate or undiluted supernatant into white 96-well assay plates. For the firefly luminescence assay, 20 μL of firefly substrate was added directly before the measurement was started (0.47 mM D-luciferin (Biosynth AG, Staad SG, Switzerland), 20 mM tricine, 2.67 mM MgSO_4_*7H_2_O, 0.1 mM EDTA*2H_2_O, 33.3 mM dithiothreitol, 0.52 mM adenosine 5′-triphosphate, 0.27 mM acetyl-coenzyme A, 5 mM NaOH, and 0.26 mM MgCO_3_*5H_2_O, in H_2_O). 

The renilla reporter gene assay was performed as described for firefly. Instead of D-luciferin as a substrate, coelenterazine (472 mM coelenterazine (Roth, Karlsruhe, Germany) stock solution in methanol) was used. The substrate was diluted in a 1:15 ratio in phosphate-buffered saline directly before use. A gaussia reporter gene assay was performed as renilla but in a 1:1500 ratio. The luminescence was measured in a Berthold technologies Tristar2S LB942 or Berthold Centro XS3 Multimode plate reader for 20 min as previously described [18,26].

For the establishment of a firefly-luciferase-based fast screening platform, 5 mL CM-glucose cultures of a firefly-containing strain were grown overnight at 28 °C. Afterward, 80 μL of the culture was transferred to white 96-well assay plates and 20 μL of firefly substrate was added as described above. The culture samples were incubated with firefly substrate for different incubation times (1, 2, 4, 8, 15, and 30 min) prior to measuring the luminescence in a Berthold Technologies Centro XS3 LB960 Microplate luminometer. After starting the determinations, the OD_600_ was measured, and the obtained values were calculated for an OD_600_ of 0.5.

### 2.4. SEAP Reporter Assay

The SEAP (human-secreted alkaline phosphatase) activity assay was performed in parallel for the supernatant and the lysed cell pellets of a harvested cell culture. A total of 100 μL of lysate and 100 μL of supernatant were pipetted into round-bottom plates for the heat inactivation of endogenous phosphatases at 65 °C for 1 h. Afterward, 80 μL of the lysate and supernatant samples was transferred into transparent 96-well flat-bottom plates and mixed with 100 μL of SEAP buffer (20 mM L-homoarginine, 1 mM MgCl_2_, and 21% (*v/v*) diethanolamine). Before the determination was started, 20 μL of 120 nM para-nitrophenylphosphate (pNPP, Sigma Aldrich, Darmstadt, Germany) was added. The absorbance was measured in a Berthold technologies Tristar2S LB942 Multimode plate reader for 2 h at 405 nm as previously described [19].

### 2.5. Fluorescence Intensity Measurements

For the fluorescence intensity measurements, 80 μL *U. maydis* culture or cell extract was transferred into a Corning 96-well flat-bottom black plate and determined in a BMG Labtech ClarioStar Multimode Plate Reader. The excitation wavelengths for eGFP and mKate2 were 470 and 588 nm, respectively, while the emission was measured at 495–535 and 605–665 nm, respectively. 

### 2.6. Induction of Filaments and Microscopy

For fluorescence analysis of filament induction, the strains were grown overnight in CM-glucose, samples were taken, and the fluorescence was measured in a plate reader Infinite M200 (Tecan Group Ltd., Männedorf, Switzerland); the excitation wavelengths for eGFP and mKate2 were 470 and 588 nm (bandwidth: 9 nm), respectively, while the emission was measured at 535 and 633 nm (bandwidth: 20 nm), respectively. For the induction of filamentous growth, the cultures were shifted to NM-glucose. Six hours later, samples of filamentous cultures were again analyzed in the plate reader as well as under the microscope. The microscopy set up was performed as described before [15,27]. In short, the Axio Observer.Z1 (Zeiss, Oberkochen, Germany) equipped with an Orca Flash4.0 camera (Hamamatsu, Japan) and objective lens Plan Apochromat (63×, NA 1.4) was used. For the excitation of eGFP (488 nm/100 mW) and mKate2 (561 nm/150 mW), a VS-LMS4 Laser Merge-System (Visitron Systems, Puchheim, Germany) combined with solid-state lasers was used. All modules of the microscope systems were controlled using the software package VisiView (Visitron). AB33 and strains for the constitutive expression of eGFP and mKate2 were used as controls.

### 2.7. Infection Assays and In Planta Luciferase Quantification

For the fluorescent analysis of *U. maydis* during plant infection, seven-day-old *Zea mays* Amadeo (KWS) seedlings were infected with fungal culture [25]. In brief, *U. maydis* strains were grown in YEPSLight medium to an OD_600_ of 1. After harvest and washing, they were resuspended to an OD_600_ of 3 and 250 µL was injected into the stem of each seedling. The plants were kept in growth chambers (CLF Plant Climatics—Gro Banks—Model TF110) with 16 h of light at 28 °C and 8 h darkness at 22 °C. Symptoms of infection were scored at 7 and 12 dpi, as described previously [7]. 

At 12 dpi, leaf tissues were harvested for luminescence measurements. For this, a 3 cm piece of the third infected leaf was cut 1 cm underneath the injection site. After determining the fresh weight, the samples were kept in a closed humid chamber and sprayed with 1 mL of luciferin substrate (the same solution as shown in chapter 2.3). After 4 h in the dark, total photon counts were detected during a 60 s period in the Berthold NightOwl LB 983. The photon count was then normalized to the weight of each leaf sample. 

## 3. Results

### 3.1. Quantitative Reporter Gene Expression

Luminescence-based and enzymatic reporter genes serve as powerful tools in synthetic biology to rapidly and quantitatively assay the expression of a gene of interest, ideally in a noninvasive manner. To generate a fast and quantitative platform for future genetic manipulations, three different luciferases originating from firefly (*Photinus pyralis*, firefly—FLuc) [28], sea pansy (*Renilla reniformis*, renilla—RLuc) [29], and the marine copepod (*Gaussia princeps*, gaussia—GLuc) [30] were customized, introduced and characterized in *U. maydis*. As an alternative approach, we implemented the human secreted alkaline phosphatase (SEAP) [31] (Figure 1A,B). These reporters have the advantage that expression in plant and animal cells can be readily read out in a growing cell culture without the need of time-consuming work-up procedures, e.g., cell lysis, either by recording the luminescence or, in case of SEAP, with a direct measurement of the supernatant after secretion. In order to generate stable reporter strains in *U. maydis*, the genes were codon-optimized, cloned under the control of the constitutive promotor P_O2tef_, and stably integrated in the *upp3* locus [10] using homologous recombination (Appendix A. Strains generated in this work). 

To evaluate whether the luciferase reporters are functionally expressed in *U. maydis*, their activities were first determined in cell lysates and whole-cell cultures. During a time-course of 20 min, the two cytosolic luciferases, FLuc and RLuc, showed high luminescence in cell lysates, and activity for the secreted GLuc was clearly detectable in whole-cell cultures (Appendix A). The luciferases showed the typical kinetics according to the enzymatic reaction: for FLuc reacting with luciferin, the luminescence was stable over 20 min (glow light behavior), RLuc reached a maximum at 5–10 min (flash light behavior), and GLuc immediately started decreasing (flash light behavior). To account for the different kinetics, we represent the data, depending on the enzymes, by taking the overall mean for FLuc [32], but the mean of only the first 10 or 3 measurements for RLuc and GLuc, respectively [17,18,26]. This reflects the linear range of the different reactions.

To verify that luminescence can be detected in a noninvasive manner, whole-cell cultures and supernatants were compared to the cell lysates (Figure 1C). All luciferase reporter strains showed high light emission upon the addition of the appropriate substrate to whole-cell cultures, whereas the wildtype background AB33 had no signal as expected. FLuc and RLuc had a >300-fold signal intensity compared to background levels, while GLuc increased 84-fold (Figure 1C). The cell lysates showed comparable light emission, but unexpectedly, the background for coelenterazine in wildtype AB33 lysates was extremely high. By contrast, SEAP activity was rather low both in the whole-cell cultures and lysates. Therefore, the luciferin-dependent FLuc was selected for further applications.

To address the sensitivity and steadiness of this reporter, the FLuc was measured in the cell culture after different time incubation periods. Almost immediately after the addition of the substrate, we detected a high activity level that remained stable over a period of 30 min (Appendix A), providing flexibility with regard to incubation time.

GLuc and SEAP are secreted in several organisms (Figure 1A) [26]. Indeed, also in *U. maydis*, the light emission of GLuc was detected in the supernatant. It was 34-fold higher than in AB33, but still at lower total levels compared to whole cells and lysates, which is indicative of the incomplete secretion of this reporter protein (Figure 1C). By contrast, SEAP activity in the supernatant was only 2-fold higher than AB33, so we excluded this reporter due to poor expression and focused on the characterization and application of the luciferases, mainly implementing FLuc and RLuc as reporters. They use different orthogonal substrates, namely luciferin and coelenterazine, which enable dual assays compatible with developing fast-screening platforms, ratiometic reporters, and *in planta* applications. 

### 3.2. Ratiometric Monitoring of Inducible Gene Expression

After having established at least two orthogonal quantitative reporters, we set to generate dual-reporter constructs for ratiometric, normalized determinations in *U. maydis* cultures. Dual reporters are widely used in synthetic biology applications and consist of one reporter for monitoring the expression of a gene of interest, and a second one that serves as an internal control, thereby yielding a direct comparison. Such an approach increases robustness and enables *in situ* normalization, rendering the quantification of gene expression relatively independent of biological sample variation, including reporter absolute expression levels, cell health, and metabolic load, among other factors [17,33]. We first characterized the expression of two strains comprising a single reporter each: RLuc under the constitutive promoter P_O2tef_ (Appendix A) and FLuc expressed under the inducible promoter P_CRG_, which is activated upon a change of carbon source from glucose to arabinose (Appendix A). Thereafter, we combined these two constructs to generate a dual reporter in the *upp3* locus of the *U. maydis* (strain sNH039, Appendix A and Figure 2A). It was shown that under inducing and noninducing conditions, the luminescence of RLuc correlated with the OD during exponential growth, which makes this protein a suitable internal normalization control (Appendix A). Importantly, only the carbon source, not the induction of the normalization reporter, had a major effect on growth (Figure 2B,C), i.e., *U. maydis* grew faster on glucose. As expected, only FLuc activity increased upon the arabinose induction (Appendix A).

The FLuc to RLuc luminescence ratio was calculated for the normalization strain over time (Figure 2D). As expected, FLuc expression increased over time relative to the constitutively expressed normalization element RLuc. This experiment describes the applicability of a normalization element that is independent of the stimulus of interest to monitor the changes in gene expression of an inducible promoter in *U. maydis* cultures.

### 3.3. Bidirectional Promoters for Bicistronic Expression in U. maydis

A major drawback in the genetic manipulation of certain host organisms, such as *U. maydis*, is the limitation of loci suitable for the insertion of foreign DNA. Additionally, the transformation process itself and the verification of a correct insertion event can be very time-consuming. Therefore, it is highly desirable to keep the number of constructs to be inserted as small as possible. The optimal solution is, therefore, to implement synthetic elements into the constructs that enable the simultaneous expression of at least two independent polypeptides from one promoter. A straightforward approach is the application of viral 2A peptides in between two open reading frames for *U. maydis* [15], leading normally to equimolar amounts of two polypeptides from one mRNA by cotranslational nascent polypeptide hydrolysis. Two alternatives to 2A sequences for the expression of at least two polypeptides from one promoter, i.e., a bicistronic construct, are bidirectional promoters (BP) (yielding two independent mRNAs) and IRES (internal ribosome entry site) sequences (two independently translated polypeptides from one mRNA) (see below). To expand the toolset for *U. maydis* further, two synthetic BPs were first engineered and tested. They comprised a central enhancer region or repeats of an enhancer, flanked by two minimal (core) promoter sequences (Figure 3A). One of the promoters was designed based on the CMV immediate early promoter enhancer flanked by two CMV minimal promoters [34]. A second variant was similarly designed; however, endogenous sequences were implemented instead of viral components. To this end, similar to the strong promoter P_OMA_ [35,36], four repeats of the prf1 enhancer were inserted in between two mfa1 minimal promoter sequences (Figure 3). Both bidirectional promoters (BPs), further called BP_CMV_ and BP_(prf)4_, were cloned for the control of expression of either mKate2-NES and eGFP-NLS, or FLuc and RLuc as reporters. The FPs tagged either to the NLS (nuclear localization signal) or NES (nuclear export sequence) were to be located in the nucleus or cytosol of the cells, respectively. Two constructs with either configuration for the reporters were engineered, i.e., each reporter either downstream or upstream. Stable transgenic strains were generated and analyzed together with the single reporter strains for their fluorescence or luminescence activity, respectively (Figure 3). 

Both BP_CMV_ strains showed comparable absolute fluorescent units (AFU) of eGFP and mKate2 (Figure 3B). In both versions of the BP_(prf)4_ strains, the relative amount of expressed protein from the coding sequence placed downstream of the enhancer (along the vector) was higher compared to the upstream construct. This resembles the results obtained by Andersen et al. [37], which showed that a minimal promoter that is aligned in the naturally occurring orientation with the enhancer is activated more efficiently than the one with the opposite orientation. The BP_(prf)4_ strains also yielded higher absolute luminescence values than the BP_CMV_, consistent with the results observed for the fluorescent reporters (Figure 3C). The engineered BP_(prf)4_ could be an interesting candidate for use in biotechnological applications, displaying strong constitutive expression but not being toxic to the cells (for the proteins tested).

### 3.4. IRES Sequences

Three commonly used IRES sequences from human poliovirus, encephalomyocarditis virus, and foot-and-mouth-disease virus were tested in *U. maydis* (Appendix A). Constructs were designed with the fluorescence proteins mKate2 and eGFP, fused to an NES and NLS, respectively, and alternatively with FLuc and RLuc as reporters. The architecture of the constructs led to the translation of mKate2-NES and RLuc being initiated by the Shine–Dalgarno sequence, whereas the downstream elements, eGFP-NLS and FLuc, were under the translational control of the IRES sequences (Appendix A).

Fluorescence determinations were performed with a plate reader on the strains grown overnight in CM-glucose. No signal could be detected above the background in the plate reader determinations (Appendix A). Therefore, we shifted the cultures to NM-glucose to induce filamentous growth and potentially boost expression. Six hours later, the samples of filamentous cultures were again analyzed in the plate reader as well as under the fluorescence microscope (Appendix A). However, IRES-containing strains still generally showed a low fluorescence compared to the P_O2tef_-driven strains, both microscopically and in the plate reader determinations. Similar results were obtained with the luciferases as the reporter. 

As they are not translated elements, the IRES sequences were not codon-optimized for *U. maydis* (which is necessary for all translated elements). The sequences potentially contain unfavorable dicodons, which might lead to the inefficient transcription of eGFP-NLS and FLuc due to premature mRNA polyadenylation, as was also previously observed for non-di-codon-optimized heterologous sequences [38]. However, if these effects were present in our experiments, at least the upstream elements mKate2-NES and RLuc should have been detectable, comparable to the case of the positive controls, as they are under the control of the same constitutive promoter. In all likelihood, the IRES might contribute to mRNA instability, resulting in a generally low expression from these constructs or, alternatively, to reduced cell viability. The latter is supported by the observation that the stronger expression of the IRES sequence seemed to be toxic and strains transformed with the P_OMA_-IRES constructs were not viable. 

In conclusion, the low expression levels achieved in strains carrying IRES constructs tested are not sufficient for biotechnological applications or synthetic expression systems.

### 3.5. Quantitative Determination of Fungal Proliferation In Planta

Finally, the tools developed here were implemented to enable the quantitative monitoring of fungal infections *in planta*. As a plant pathogen, *U. maydis* colonizes its host maize and subsequently induces tumor formation. Seedling infection assays are standard approaches to address virulence phenotypes in genetically engineered strains. However, the readout is symptom scoring [7], which remains semiquantitative. To assess fungal biomass, the infected material can be harvested and either stained or its biomass can be quantified using qPCR on genomic DNA [39]. Fluorescent proteins are difficult to use in quantification due to the high plant background. Therefore, we implemented the FLuc reporter in infection assays to be able to quantitatively and spatially monitor fungal growth and infection. 

The experimental set up was based on the haploid strain SG200, which is solopathogenic and can infect the plant in the absence of a mating partner due to autoactive mating loci [7]. It is expected to show full virulence. As a negative control, we included AB33-FLuc, since this haploid strain is lacking a mating partner for infection and is, therefore, avirulent when injected alone. As an additional control, SG200-pit1∆-FLuc was generated. The virulence factor Pit1 is required for tumor formation. While the deletion mutant can penetrate the host plant, proliferation in planta remains locally restricted [23]. Hence, the SG200-pit1∆-FLuc strain showed an intermediate colonization rate.

Using these three reporter strains, in comparison to the parental strains (considered wildtype here), we first confirmed that the FLuc does not influence virulence. As expected, symptom development was indistinguishable from the respective progenitor strain (Figure 4A): AB33 and AB33-FLuc were avirulent, SG200 and SG200-FLuc were fully virulent and induced large tumors, whereas pit1∆ and pit1∆-FLuc showed an intermediate phenotype with only ~50% tumors. 

Next, the plants were sprayed with luciferin and cumulative luminescence was measured. Tumors that were filled with fungal hyphae (SG200-FLuc) emitted a strong light signal, whereas healthy plants (mock) or tumors of strains lacking the FLuc (SG200 and SG200-pit1∆) did not show any background signal (Figure 4B,C). Interestingly, leaves with chlorosis sometimes had high luminescence (SG200-pit1∆), suggesting that a fungal mycelium is growing in these leaves, whereas in other cases, there was no luminescence suggesting no fungal growth (AB33-FLuc). By contrast, leaves that build anthocyanin generally did not show luminescence, suggesting that the plant immune system successfully fought off the fungus (Figure 4B). Hence, the correlation of symptom development and luminescence within one leaf can even give an insight into where the fungus is arrested. 

Overall, the SG200-FLuc strain is now a valuable genetic background for the analysis of genes related to virulence. It has the potential to become as widely used in plant–microbe interactions as the Pst-LUX bacterial reporter strain. For example, it can be used as a parental strain for mutant analysis in *U. maydis* by the smut fungal community, as well as in testing maize lines for susceptibility towards smut fungal infection. In addition, this example illustrates that bioluminescence-dependent pathogen quantification can be adapted to pathogens beyond the bacterial kingdom. 

## 4. Discussion

In summary, here, we characterized three luciferases as useful reporters for *U. maydis* and identified FLuc as the most applicable. The high enzymatic activity and low background of FLuc in the whole culture enabled us to establish a FLuc-based fast-screening platform that does not require lysis and the direct measurement of the reporter gene activity in culture, e.g., for the implementation of a rapid-screening platform for gene expression in *U. maydis*. Besides applications in cell cultures, FLuc can also be used to monitor fungal proliferation in planta during infection. Similar to the bacterial *Pst*-LUX strains, this new strain has the potential to become a new standard strain in the community.

Moreover, the bidirectional promoters implemented here provide a means to engineer polycistronic expression in one construct/insertion. They increase the efficiency of synthetic module integration, which will be a powerful tool to facilitate multigene expression in *U. maydis,* as also needed for the implementation of more complex synthetic switches (chemically inducible, optogenetics, etc.). The available approach which previously existed in the widely used strain AB33 was the implementation of two independent nitrate-inducible Pnar promoters, placed in bidirectional orientation and interspersed by an antibiotic resistance cassette, for the control of b-gene expression and, thereby, filament induction [8]. However, this is not a single promoter as is the case for the synthetic bidirectional promoters established here which are much more compact and customizable. In addition, as a readout, we used the dual-luciferase (FLuc and RLuc) assay system to monitor the ratiometric activity in *U. maydis*. In the future, this combination can greatly facilitate the non-invasive monitoring of inducible gene expression under varying conditions in culture and during infection. Upon establishment in *U. maydis*, these constructs might also be transferable to other fungi such as *T. thlaspeos* [40], or even to other species beyond the smut fungi.

## Figures and Tables

**Figure 1 jof-09-00480-f001:**
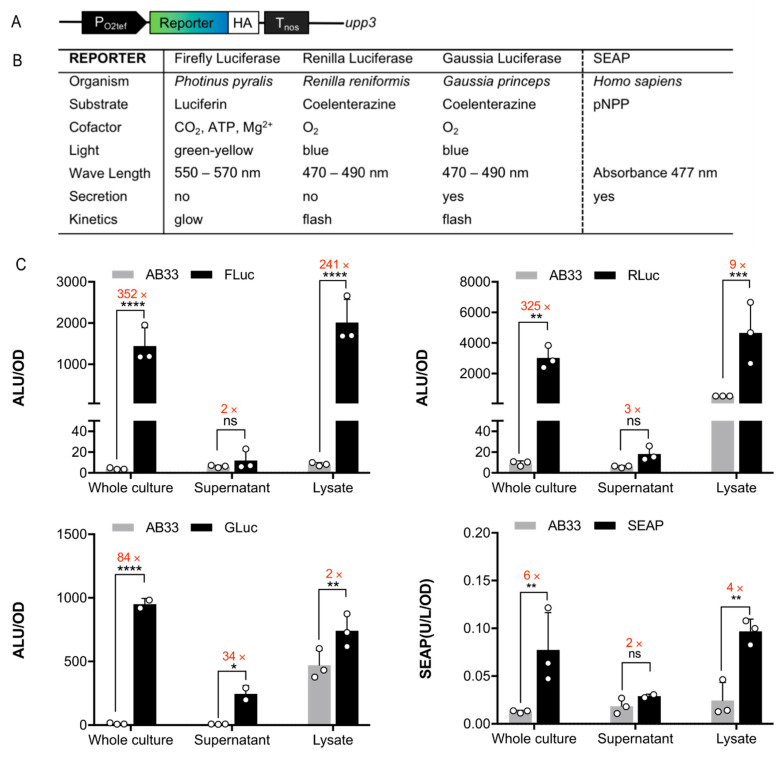
Quantitative reporter gene expression in *U. maydis*. (**A**) Molecular configuration of the constructs transformed for stable integration into the *upp3* locus of the AB33 wildtype strain. The reporter genes were expressed under a constitutive promoter (P_O2tef_). HA, synthetic peptide tag for Western blot analysis; T_nos_, *nos* terminator. (**B**) Description and parameters of the selected reporter genes. (**C**) Quantitative characterization of reporter gene activities. Whole-cell culture, supernatant, and lysate of 80 µL culture were analyzed in all reporter strains, and the original AB33 WT strain served as a negative control. *Upper left*, FLuc luminescence after addition of D-luciferin; *upper right,* RLuc luminescence after addition of 1:15 diluted coelenterazine in PBS; *lower left,* GLuc luminescence after addition of 1:1500 diluted coelenterazine in PBS; *lower right,* SEAP activity after addition of 4-nitrophenyl phosphate (pNPP). Luminescence is given as absolute luminescence units (ALU) and normalized to OD_600_ = 0.5. Error bars, s.d. (n = 3). *, *p* < 0.05; **, *p* < 0.001; ***, *p* < 0.0002; ****, *p* < 0.0001; ns, not significant (one-way ANOVA).

**Figure 2 jof-09-00480-f002:**
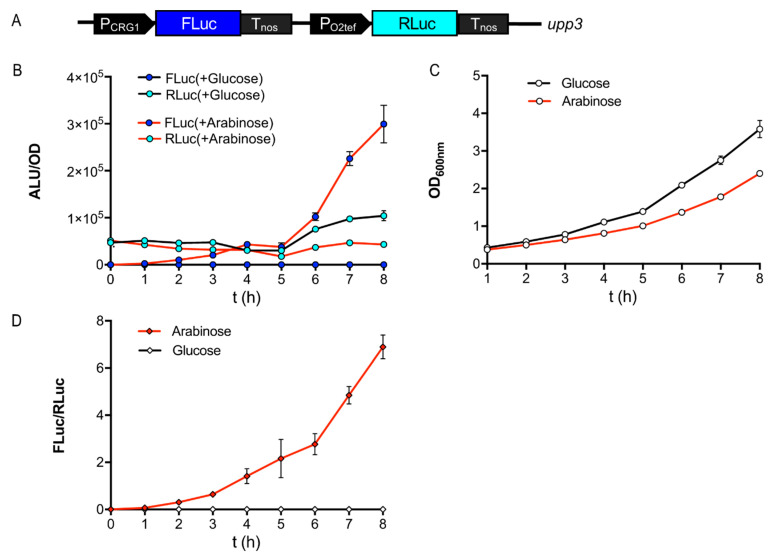
Ratiometric determination of inducible gene expression in *U. maydis*. (**A**) Structure of the FLuc/RLuc dual-reporter construct transformed into the *upp3* locus of AB33 for generation of the ratiometric strain (sNH039; for detailed information, see Appendix A). The normalization strain expressing FLuc under control of the arabinose-inducible promotor (P_CRG1_) and RLuc under control of the constitutive promotor PO2tef was used to calculate the FLuc/RLuc ratio for cultures in CM-glucose or arabinose medium. (**B**) Absolute luminescence of whole-cell lysates from 2 mL culture of the strains (normalized to an OD600 = 0.5). Cultures were grown overnight in CM-glucose and (i) kept in CM-glucose (black line), or (ii) shifted to CM-arabinose at time-point 0 for determinations (red line). Samples were taken for lysates every hour and analyzed for FLuc (blue dots) and RLuc (cyan dots) luminescence. (**C**) Growth of the transformed strains in CM-glucose or arabinose (in panel A, B) over a time period of 8 h, followed as OD_600_. (**D**) FLuc to RLuc ratio of the normalization strain over time. Error bars represent the SEM of this individual experiment with n = 3.

**Figure 3 jof-09-00480-f003:**
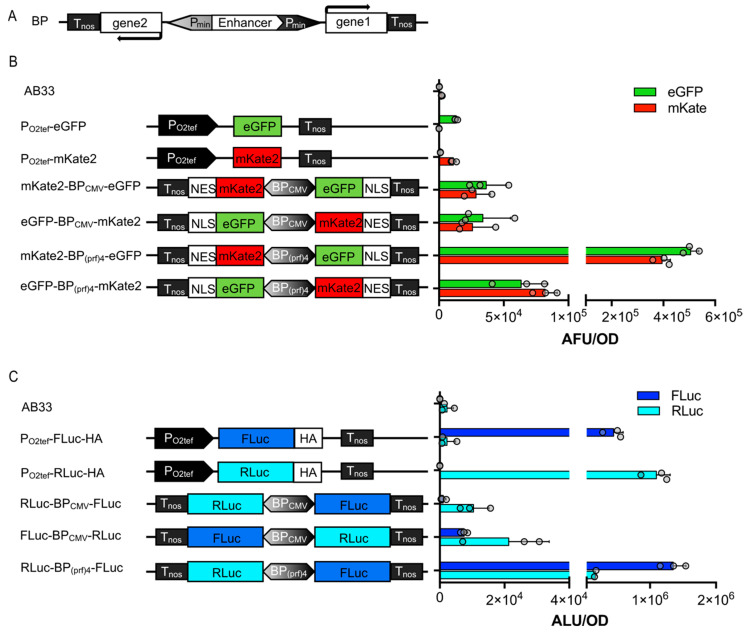
Establishment of bidirectional promoters in *U. maydis*. (**A**) Configuration of constructs with bidirectional promoters that have been transformed into the *upp3* locus of AB33. The minimal promoter that is located upstream of the enhancer is on the light-gray end of the bidirectional promoter (grayscale gradient). (**B**) Fluorescence intensity of strains determined in cultures of an OD_600_ = 0.5. (**C**) Absolute luminescence of cell lysates from 2 mL cultures; both controls and BP strains were adjusted to OD_600_ = 0.5. Error bars represent the SEM of the individual experiments with n = 3.

**Figure 4 jof-09-00480-f004:**
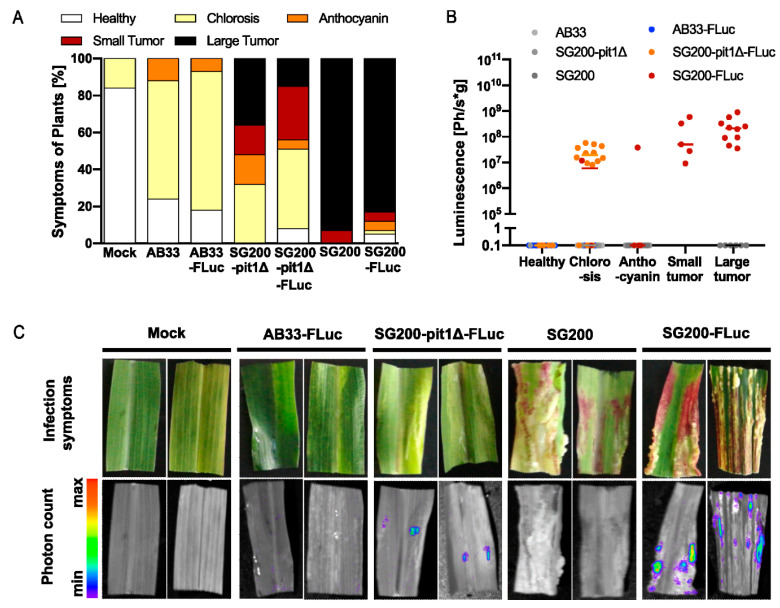
Quantitative monitoring of infections symptoms in maize infection assays. (**A**) Symptom scoring of infected maize plants at 12 dpi. As expected, pathogenicity increased from AB33 (nonvirulent) via SG200-pit1Δ (virulent up to entry) to SG200 (fully virulent). (**B**) Luminescence-based analysis of fungal proliferation in plant tissue. Symptoms of each sample were scored and are indicated by colored dots. The non-pathogenic strain AB33-Fluc showed background luminescence, as the fungus was not able to proliferate and produce biomass in planta. SG200-pit1Δ-Fluc infection resulted in luminescence corresponding to fungal entry, but lack of proliferation in the host plant. For SG200-Fluc, high photon counts were detected, which corresponds to the fully virulent phenotype of the strain. (**C**) Pictures taken with the NightOwl device during photon count measurement. Intense luminescence signals were found in all tissues with high amounts of fungal biomass, mostly in tumors filled with fungal hyphae.

## Data Availability

All data generated or analyzed during this study are included in this published article and its supplementary information.

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
