# Peer review of "Engineering and Implementation of Synthetic Molecular Tools in the Basidiomycete Fungus Ustilago maydis"

_jof, 2023, doi:10.3390/jof9040480_

Round 1

Reviewer 1 Report

-          The work proposed “Engineering and implementation of synthetic molecular tools in the basidiomycete fungus Ustilago maydis” by the authors Nicole Heucken, Kun Tang, Lisa Hüsemann, Natascha Heßler, Kira Müntjes, Michael Feldbrügge, Vera Göhre, and Matias D. Zurbriggen may be of interest to readers who are researchers in the field of biology molecular, special those people who are developing synthetic tools used as a fast-screening platform for reporter gene expression, applicable to in vitro and in vivo detection. Therefore, I consider that this work may be of interest to be published.

-          However, I suggest only a pair of corrections throughout the text.

-          1- The abstract and introduction should be written in the third person. Some phrases shout be modified. The manuscript is highlighted in yellow. Please check all text

-          2- Section: 2. Materials and Methods 2.1. Plasmid generation

-          -              On-Line 66 – It is written: “Plasmids and oligonucleotides used here are listed and described in Table S1 and Table S2”   please, could you correct the names of the tables because the oligonucleotides are in Table S3 in the supplementary material

Author Response

Thank you very much for the positive feedback and the suggestions for improvement.

"1- The abstract and introduction should be written in the third person. Some phrases shout be modified. The manuscript is highlighted in yellow. Please check all text"

We have now avoided the use of "we" and changed the sentences to the third person or passive voice as suggested.

"2- Section: 2. Materials and Methods 2.1. Plasmid generation

On-Line 66 – It is written: “Plasmids and oligonucleotides used here are listed and described in Table S1 and Table S2”   please, could you correct the names of the tables because the oligonucleotides are in Table S3 in the supplementary material"

We have changed the order of the tables in the supplementary figure, so that they are now numbered correctly in the order of appearance in the text.

Reviewer 2 Report

The study evaluated about "Engineering and implementation of synthetic molecular tools 2 in the basidiomycete fungus Ustilago maydis". 

This original paper was prepared well with all parts of the manuscript very deeply and detail.

Finally, It is very well-marked that this paper is acceptable in the present form and useful for publish in this journal.

Author Response

Thank you very much for the positive feedback.

Reviewer 3 Report

In this manuscript , the fungus Ustilago maydis is used as a model. The expression of different luciferase genes under the control of bidirectional or bicisitronic promoters was constructed and evaluated. The FLuc luciferase gene was the most efficient in U. maydis, which is a novel alternative in the studies of this fungus, allowing the quantification of the reporter protein in different conditions, such as the interaction of the fungus with the plant, which cannot be adequately performed with other reporter proteins, such as fluorescent proteins.

It's necessary include synthetic molecular tools described for U. maydis in the introduction and discussion, such as the CRISPR–Cas system. The addition of references of molecular tools will allow a better reference in the readers over proposal of the molecular tools of this manuscript.

Author Response

Thank you very much for the positive feedback and the suggestion.

"It's necessary include synthetic molecular tools described for U. maydis in the introduction and discussion, such as the CRISPR–Cas system. The addition of references of molecular tools will allow a better reference in the readers over proposal of the molecular tools of this manuscript."

We have included the CRIPR-Cas system as described in Schuster 2016 in the introduction (Line 41/42). This now nicely shows that different techniques are available for genetic modification of U. maydis.